# Effect of Snow Cover on Spring Soil Moisture Content in Key Agricultural Areas of Northeast China

**Mingxi Pan [1,2], Fang Zhao [3,4], Jingyan Ma [1,*], Lijuan Zhang [1], Jinping Qu [5], Liling Xu [2] and Yao Li [6]**

1   Heilongjiang Province Key Laboratory of Geographical Environment Monitoring and Spatial Information Service in Cold Regions, Harbin Normal University, Harbin 150025, China; mhjpmx@126.com (M.P.); zhlj@hrbnu.edu.cn (L.Z.)
2   Mohe Meteorological Bureau, Mohe 165300, China; mhxll@126.com
3   Harbin Meteorological Bureau, Harbin 150036, China; zhaofang213022@163.com
4   Heilongjiang Province Institute of Meteorological Sciences, Harbin 150036, China
5   Tieli Meteorological Bureau, Tieli 152500, China; ziyu-571@126.com
6   Daxing'anling District Meteorological Bureau, Daxing'anling 165000, China; lz4818@163.com
*   Correspondence: 18686818800@163.com

**Abstract:** As an important source of soil moisture content during spring in high-latitude areas, snow cover affects the occurrence of spring drought and crop yield and quality. There has not been sufficient research on the effect of winter snow cover on spring soil moisture content. This paper focuses on the main agricultural areas of Northeast China—the Songnen Plain and the Sanjiang Plain. Using meteorological data of both spring soil moisture content and snow cover at 19 agricultural meteorological stations from 1983 to 2019, the effect of snow cover on spring soil moisture content in the Sanjiang Plain and Songnen Plain is studied by variance analysis, spatial analysis, and correlation analysis. The results show that: (1) Compared to the Sanjiang Plain, the Songnen Plain has a significantly lower content of soil moisture at the surface (0–10 cm) and deep layer (10–20 cm, 20–30 cm) during the entire spring and every month of spring ($p < 0.05$), and a greater interannual variation of soil moisture. (2) Snow cover has a significant effect on spring soil moisture in the Songnen Plain, but not as much as one in the Sanjiang Plain. For the Songnen Plain, snow-cover duration and the snow-cover onset date has a lasting influence on spring soil moisture until May, which can extend to as deep as 20–30 cm. As months go by, its influence on shallow-layer soil gradually wears off. Maximum snow depth and the snow-cover end date only influence the April surface soil. (3) Snow cover has a strong effect on soil moisture conservation in more arid areas. Delayed snow-cover onset date, earlier snow-cover end date, and significantly shortened snow-cover duration all contribute to a spring drought soil condition in the Songnen Plain.

**Keywords:** snow cover; spring soil moisture; impact mechanism; Songnen Plain; Sanjiang Plain

## 1. Introduction

Snow is the most active constituent of the cryosphere [1]. In high-latitude areas, water is released in the form of melted snow in just a few days [2] and comprises an important source of soil moisture in spring [3,4]. At the same time, snow cover also reduces the change in soil moisture content and temperature by hindering the energy exchange between soil moisture, temperature, and the environment [5], thus conserving soil moisture. However, due to global warming, the area of snow cover has dropped significantly in the northern hemisphere in the past few decades [6,7]. Snow duration in the northern hemisphere decreases at a rate of 5.3 d/10 a [8]. Particularly prominent changes in snow cover are observed in spring. Snow cover area has seen a significant reduction during spring in the northern hemisphere [9], and snow starts to melt at a significantly earlier time in Eurasia during spring [10–13]. The decline in snow reserves and the rapid, earlier disappearance of snow cover, which causes the spring warming and soil aridness, have been the center

of widespread concern in the scientific community, and they have become an important influencing factor on the degree and duration of soil aridity in spring [14,15]. Researchers both in China and overseas have studied the impact of snow cover on soil moisture through the one-point method, actual observations, remote sensing, and simulation. These studies are mainly carried out from three aspects. First, some researchers have studied the effect of snow cover on the level of spring soil moisture. For example, Shinoda [16] studied the relationship between snowmelt and soil moisture in Central Eurasia with data collected at meteorological stations and found that greater annual maximum snow depth and delayed snow-cover end date correspond to higher level soil moisture, and vice versa. Ren et al. [17] studied the effect of snowmelt on soil water and heat conditions. Their results indicate that snowmelt significantly increases the water content of shallow-layer soil. Niu et al. [18] conducted field experiments to observe the change in soil moisture content as snow melting takes place, and they concluded that snowmelt infiltration acts to a certain extent to replenish the water content of the soil. Qi [19] found through simulation that without snow, the soil moisture level in Northeast China will drop at least 20% in the March to May period. Second, some researchers study the effect that the duration of the impact snowmelt has on soil moisture. For example, Douville [20] performed simulations with the Meteor-France GCM and suggested that the effect of spring snowmelt on soil moisture can last until summer. McNamara [21] simulated and analyzed the variational characteristics of soil moisture using observed data and modeling, and concluded that snowmelt, rainfall, and evaporation jointly drive the water and heat balance in soil moisture during late spring. Zhang et al. [22] reported that snow cover has an impact on soil moisture mainly as it melts. Third, some researchers have investigated how the depth of snowmelt affects soil moisture content. For example, Jan [23] proposed that 200 mm of snowmelt water had a very small effect on spring soil moisture content for soil layers below the 90-cm horizon. Zhang [10] used soil moisture data in conjunction with snow cover stage to analyze the influence of snowmelt on the humidity of seasonal frozen soil; the research shows that the maximum depth of snow cover that influences soil moisture content is 20 cm. Flerchinger et al. [24] conducted experiments to simulate the physical process of groundwater recharge by infiltrated snowmelt. It can be seen that snow cover has a certain replenishment effect on spring soil moisture. Greater snow-cover depth and longer snow-cover duration tend to have a more significant impact on soil moisture content. However, there is no consistent conclusion on that how much time the effect of snow cover can last and how deep the snow cover can affect. Different researchers have drawn different conclusions, which may be related to differences of research areas, but there is no further study. In addition, it is no certain answer about how much spring soil moisture is from snow cover. All these problems require further research, which are also the scientific problems that this study wants to solve.

As the largest crop production site in China, Heilongjiang Province often suffers from particularly serious spring droughts. This affects crop yield and quality. However, research on the factors influencing spring soil moisture in Heilongjiang Province has mostly been focused on temperature and precipitation [25–28]. As a region with a stable amount of snow cover, the effect of winter snow cover on soil moisture conservation in spring has not been sufficiently studied. In this paper, two agricultural bases of Heilongjiang Province, i.e., the Songnen Plain and Sanjiang Plain, which have clearly different soil moisture contents, are selected as the study sites. Using the soil moisture and meteorological data (1983–2019) from 19 agricultural meteorological stations, the impact of snow cover on spring soil moisture is analyzed for both sites. The objectives of present study were to (1) determine the variance analysis of two agricultural bases in Heilongjiang Province. (2) discuss the influence of snow cover on spring soil moisture content (3) reveal the contribution by snow-cover conditions on spring soil moisture. The results of this study form the scientific basis for the early warning of spring drought, the development of more efficient irrigation schemes, and crop yield prediction.

## 2. Materials Methods

### 2.1. Study Sites

The two main agricultural bases of Heilongjiang Province, the Songnen Plain and Sanjiang Plain (Figure 1), were chosen as the study sites. The Songnen Plain is in the western part of Heilongjiang Province. With a crop production area of 5.35 million hm$^2$, this region makes up 52% of the total crop production area of the province. It has a high-latitude continental monsoon climate, with high temperature and frequent rain in summer, severe cold and little rain in winter, and a short spring and autumn. Temperature in this region tends to have sharp fluctuations. The summer temperature is high, with high temperature and frequent rain occurring in the same season. The Sanjiang Plain is in the northeastern part of Heilongjiang Province, and it has a cultivated land area of 2.696 million hm$^2$. It has a temperate humid and semi-humid monsoon climate, with large annual temperature variation and abundant rainfall. Heilongjiang Province is one of the regions in China that has a stable amount of snow cover. The winter snow-cover period is primarily from November to April of the next year. During this time, the entire study area is covered with snow, with the maximum snow depth occurring in February. The snow begins to melt in March and completely melts in mid-April. Crop cultivation typically starts in May. Main food crops grown in this area include rice, wheat, corn, and soybean.

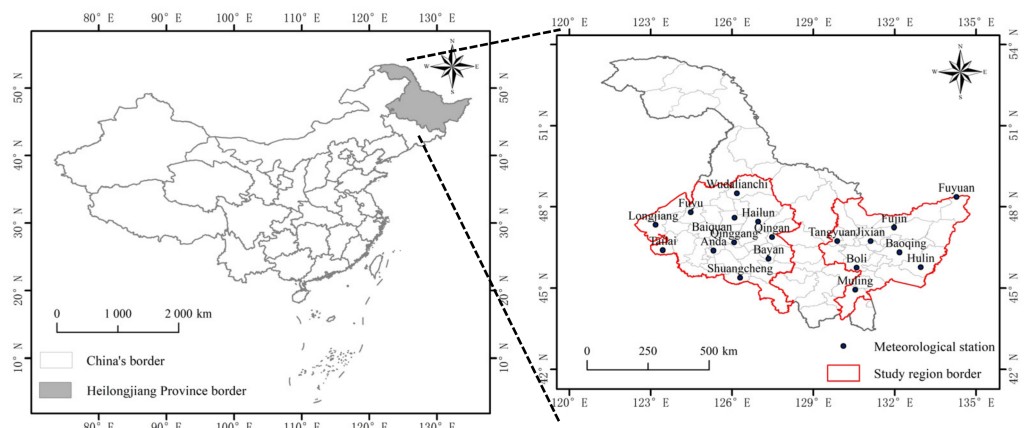

**Figure 1.** The study region and distribution of 19 meteorological stations in Heilongjiang Province, China.

### 2.2. Data Sources

Soil moisture data: spring soil moisture data for each ten days (March–May) from 19 agricultural meteorological stations (Figure 1). The time period covered spans 1983–2019. Observations were made on three soil depth layers (0–10 cm, 10–20 cm, and 20–30 cm). The data were obtained from the Meteorological Department of Heilongjiang Province. There are 31 agricultural meteorological stations in Heilongjiang Province with soil moisture data available. Provincial-wide recording of soil moisture started in 1980, but for each station, the exact commencement time is different, ranging from 1980 to 1987. To ensure continuity of data, soil moisture data from 19 agricultural meteorological stations spanning 1983–2019 were used in this study, with 11 stations on the Songnen Plain and 8 stations on the Sanjiang Plain. The data collected adequately reflect the climate characteristics of the main agricultural sites of Heilongjiang Province. Soil layers at the 0–30 cm depth range were selected for analysis, as the past study has shown that this depth interval was found to show the greatest variability in soil effective water content, and it is the cultivated layer of the soil. Soil moisture observations were all made in the observation field of agricultural meteorological stations, on grassland. No irrigation activity was carried out during the study period. The stations are located on land with different soil types, with black loam, sandy loam, clay loam, loam, yellow sandy soil, and dark brown loam being the most common ones. To eliminate the impact of soil type on soil moisture content, relative soil humidity was used to characterize soil moisture content. Relative soil humidity (RSH) is

defined as the ratio between percent soil moisture (weight basis) and field capacity, and it is expressed as a percentage (%). Field capacity is the greatest amount of water that can be stably held by soil, and it is a constant value. As such, relative soil humidity can be used to characterize soil moisture and compare the degree of soil dryness and wetness between different regions. Using this value as an index, the degree of drought is defined on the following scale: severe drought: RSH ≤ 40%; moderate drought: 40% < RSH ≤ 50%; mild drought: 50% < RSH ≤ 60%; suitable for agriculture: 60% < RSH ≤ 90%; waterlogged: RSH > 90%.

Meteorological parameters: the following daily meteorological data were used: temperature, total precipitation, previous autumn precipitation, average surface temperature, average wind speed, sunshine duration, snow-cover duration, maximum snow depth, snow onset date, and snow-cover end date. Temperature, precipitation, surface temperature, wind speed, and sunshine duration were all taken in March–May of 1983–2019. The previous autumn precipitation is the amount of rainfall in September–November of the previous year. Snow-cover duration is the number of days with snow cover from August of the previous year to July of the present year. Maximum snow depth is the greatest depth of snow recorded from August of the previous year to July of the present year.

### 2.3. Data Analysis

### 2.3.1. Variance Analysis

The soil moisture content was analyzed by one-way analysis of variance (ANOVA). These statistical tests determine whether the differences in the soil moisture of the Songnen Plain and Sanjiang Plain were significant. Duncan's shortest significant range method was used to test the differences in the soil moisture of the Songnen Plain and Sanjiang Plain and the level of significance. First, the sum of squares for the deviations was obtained for each set of data. The statistical independence of the data among each treatment was then tested. The probability of the events was given when the statistic was greater than the F value, i.e., $p\{>F\} = p$. When $p < 0.01$, the difference was considered extremely significant; when $0.01 < p < 0.05$, the difference was considered significant; when $p > 0.05$, the difference was considered not significant.

### 2.3.2. Spatial Assessment Method

The spatial distribution characteristics of soil moisture content of the Songnen Plain and Sanjiang Plain were statistically calculated via the ArcGIS Grid module and Spatial Analyst module. The Kriging interpolation method is employed to analyze the spatial distribution of soil moisture content. Kriging is a regression algorithm for spatial modeling and prediction (interpolation) of random processes/random fields based on covariance functions.

### 2.3.3. Correlation Analysis

Correlation analysis is statistical method used to discover if there is a relationship between two variables/datasets, and the relatedness and negative/positive correlation of this relationship. The Pearson correlation method is adopted in this paper to analyze the relationship between soil moisture content and climate indicators quantitatively. In each pair, the Pearson's correlation coefficient ($r$) is calculated as:

$$r = \frac{\sum_{i=1}^{n}(x_i - \overline{x})(y_i - \overline{y})}{\sqrt{\sum_{i=1}^{n}(x_i - \overline{x})^2}\sqrt{\sum_{i=1}^{n}(y_i - \overline{y})^2}} \tag{1}$$

where $\overline{x}$ represents the soil moisture content, and $\overline{y}$ represents meteorological parameters.

### 2.3.4. Percentage Contribution

To quantitatively study the long-term percentage contribution by snow cover and other meteorological parameters on the spring soil moisture content at different depths, multiple linear regression was performed on soil moisture content at all depths and meteorological

parameters of significant relevance for different spring months. The following standardized regression equation was obtained:

$$S_i = \sum a_j \times M_j \tag{2}$$

where $S_i$ is the soil moisture content at different depths across spring months; $M_j$ is the meteorological parameter significantly related to $S_i$; and $a_j$ is the normalization constant of $M_j$.

Using Equation (3), the percent contribution by each meteorological parameter on spring soil moisture across different soil depths is calculated:

$$ConM_j = \frac{a_j}{\sum_{i=1}^{n} a_i} \times R^2 \times 100\% \tag{3}$$

where $ConM_j$ is the percent contribution by $M_j$ to spring soil moisture; $a$ is the normalization constant for each meteorological parameter; $n$ is the number of meteorological parameters; and $R^2$ is the goodness of fit of the normalized regression equation.

## 3. Results

*3.1. Comparison of Spring Soil Moisture Content for the Songnen and Sanjiang Plains for 1983–2019*

3.1.1. Comparison of Spring Soil Moisture Content at the 0–30 cm Soil Depth

Figure 2 shows the spatial distribution of spring soil moisture on both the Songnen and Sanjiang Plains for 1983–2019. The Sanjiang Plain has significantly higher soil moisture content than the Songnen Plain. Spring soil moisture content varies between 69.35% and 97.44% for the Songnen Plain during this time period, and the average value is 81.39%. The coefficient of interannual variation is 0.09, and the coefficient of spatial variation is 0.072. Spring soil moisture content varies between 81.44% and 110.0% for the Songnen Plain during this time period, and the average value is 92.37%. The coefficient of interannual variation is 0.07 and the coefficient of spatial variation is 0.101. These results show that the Songnen Plain has a greater interannual variation of soil moisture than the Sanjiang Plain, whereas the intra-region variation is smaller. Variance analysis shows that there is a significant difference in spring soil moisture content between the Songnen Plain and the Sanjiang Plain, with that of the former significantly lower than the latter ($p < 0.01$).

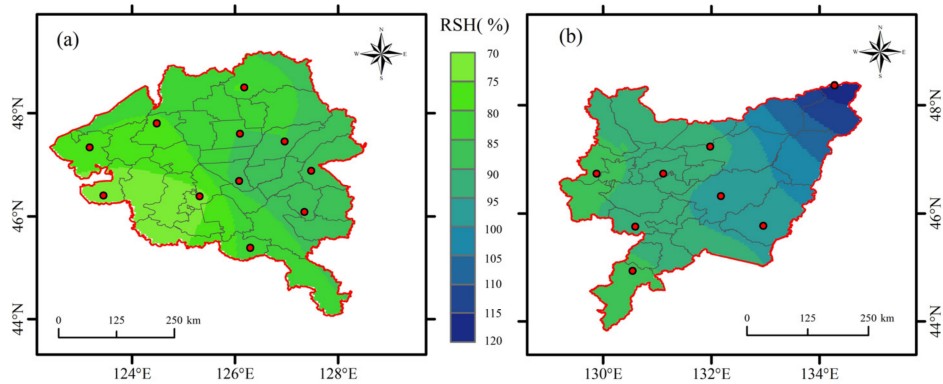

**Figure 2.** The spatial distribution of spring soil moisture at the 0–30 cm soil depth on both the Songnen and Sanjiang Plains for 1983–2019. ((**a**): Songnen Plain, (**b**): Sanjiang Plain).

Figure 3 shows the spatial distribution of spring soil moisture at the 0–30 cm depth in different month on the Songnen Plain and Sanjiang Plain for 1983–2019. On the Songnen Plain, the ranges of soil moisture content are 64.38–92.59%, 72.63–91.01%, and 72.30–84.73%, in March–May, respectively. On the Sanjiang Plain, the values are 89.05–129.70%, 84.96–122.54%, and 83.17–102.20% in March–May, respectively. It is obvious that the Sanjiang Plain has

higher soil moisture content than the Songnen Plain. Variance analysis shows that there is a significant difference in spring soil moisture content between the Songnen Plain and the Sanjiang Plain in each month, with that of the former significantly lower than the latter ($p < 0.01$).

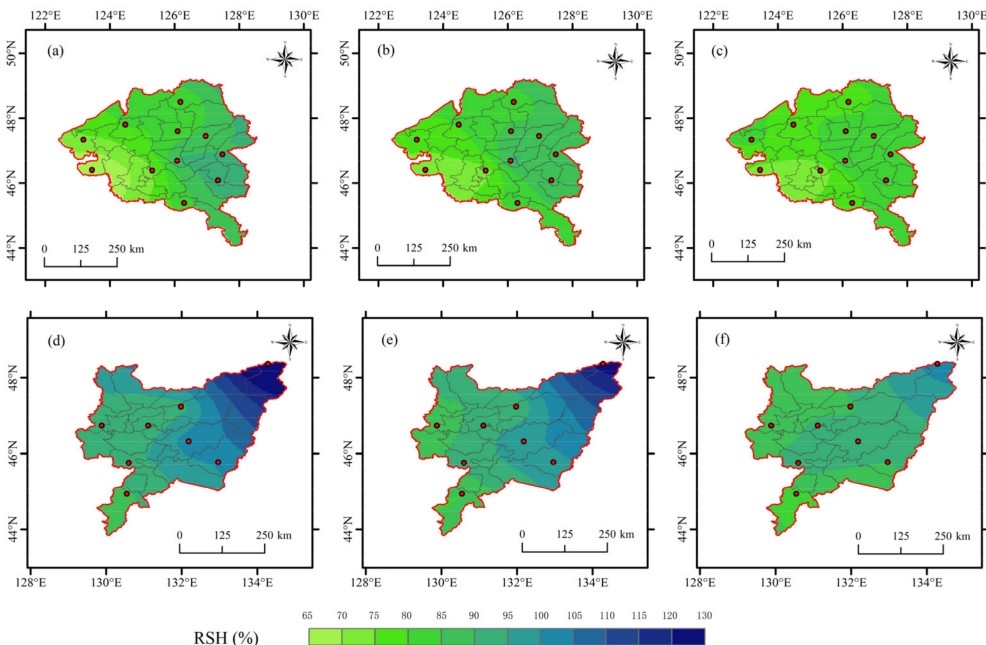

**Figure 3.** The spatial distribution of spring soil moisture at the 0–30 cm soil depth in different month on both the Songnen and Sanjiang Plains for 1983–2019. ((**a**): Songnen Plain in March, (**b**): Songnen Plain in April, (**c**): Songnen Plain in May, (**d**): Sanjiang Plain in March, (**e**): Sanjiang Plain in April, (**f**): Sanjiang Plain in May).

### 3.1.2. Comparison of Spring Soil Moisture in Across Soil Layers

Figure 4 shows the spatial distribution of spring soil moisture at different depths on the Songnen Plain and Sanjiang Plain for 1983–2019. On the Songnen Plain, the ranges of soil moisture content are 51.90–82.30%, 70.83–91.27%, and 76.34–98.69, at 0–10 cm, 10–20 cm, and 20–30 cm depths, respectively. On the Sanjiang Plain, the values are 82.94–118.37%, 88.21–130.59%, and 91.81–133.29% for 0–10 cm, 10–20 cm, and 20–30 cm depths, respectively. The soil moisture content of the Songnen Plain is lower than that of the Sanjiang Plain at each corresponding depth. Variance analysis further shows that significant differences exist in spring soil moisture at the depth ranges of 0–10 cm, 10–20 cm, and 20–30 cm between the Songnen Plain and the Sanjiang Plain. For the surface soil layer of the Songnen Plain (0–10 cm), the lowest level of soil moisture is observed in Tailai, indicating a state of mild drought, while soil conditions at other stations are suitable for farming. Conversely, 37% of the stations in the Sanjiang Plain feature waterlogged soil. These stations are primarily located in the Boli–Baoqing–Fuyuan region, while the soil moisture level at other stations is suitable for agriculture.

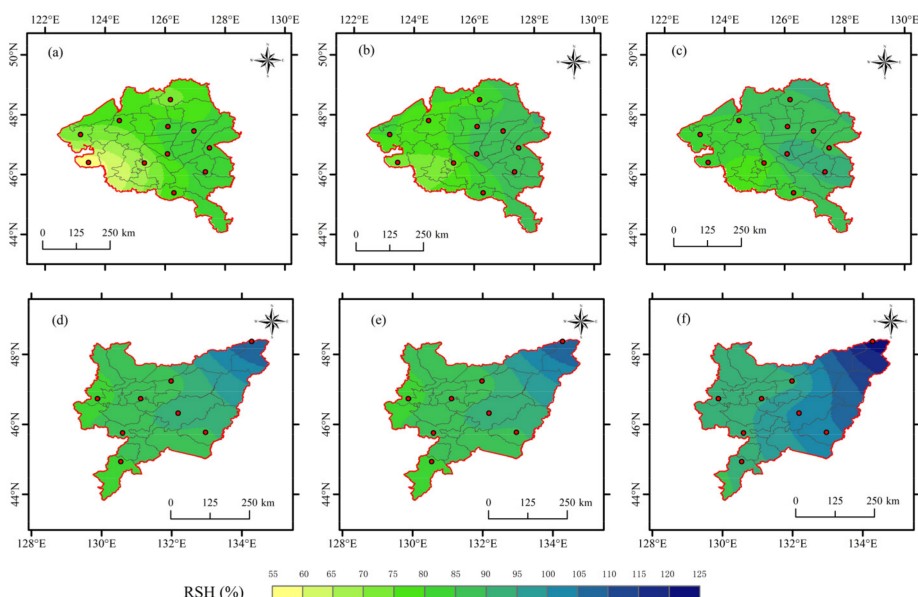

**Figure 4.** The spatial distribution of spring soil moisture at different depths on the Songnen Plain and Sanjiang Plain for 1983–2019. ((**a**): 0–10 cm of Songnen Plain, (**b**): 10–20 cmof Songnen Plain, (**c**): 20–30 cm of Songnen Plain, (**d**): 0–10 cm of Sanjiang Plain, (**e**): 10–20 cm of Sanjiang Plain, (**f**): 20–30 cm of Sanjiang Plain).

Figure 5 shows a comparison of spring soil moisture at different soil depths in each month for both the Songnen Plain and the Sanjiang Plain from 1983 to 2019. Variance analysis results showed that compared to the Sanjiang Plain, the Songnen Plain has significantly lower soil moisture content across various soil depths in spring months ($p < 0.05$). In some areas of the Songnen Plain, the 0–10 cm soil layer experiences drought of varying degrees in the March–May period (RSH < 60%, indicating mild drought), while on the Sanjiang Plain, the soil moisture content across different soil layers shows a mostly waterlogged state for the spring months. Compared to the Sanjiang Plain, the soil moisture content of the Songnen Plain shows greater interannual variation during spring months, with March being the largest in variation range (nearly 15%).

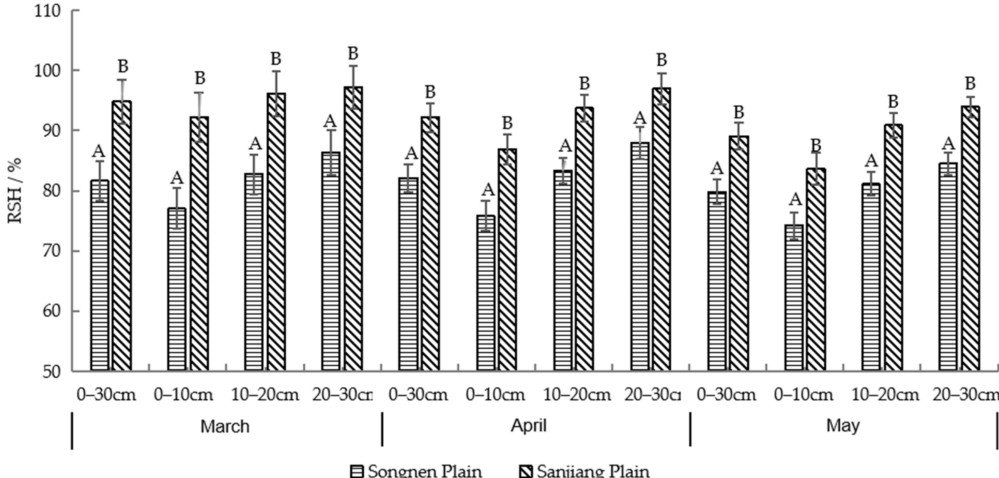

**Figure 5.** The spring soil moisture at different soil depths in each month for both the Songnen Plain and the Sanjiang Plain from 1983 to 2019. Note: "A", "B": Variance analysis result between the Songnen Plain and the Sanjiang Plain.

The above spring soil moisture distribution results show that the Songnen Plain has a significantly lower spring soil moisture content than the Sanjiang Plain during the entire spring season, each month of the season, across the entire soil layer, and at every depth.

### 3.2. Factors Influencing the Spring Soil Moisture Content of the Songnen Plain and Sanjiang Plain

Numerous studies have shown that soil moisture is closely related to meteorological factors. Taking into consideration the climate characteristics of Heilongjiang Province, 10 meteorological factors, including snow-cover conditions, were selected to perform correlation analysis with spring soil moisture at different depths and for each month of spring at different depths. The 10 factors are: daily average temperature (T), total precipitation (P), surface temperature (ST), daily average wind speed (WS), daily average sunshine duration (SD), precipitation in the previous autumn (PPA), snow-cover duration (SCD), maximum snow depth (MSD), snow-cover end date (SED), and snow-cover onset date (SOD).

### 3.2.1. Factors Affecting Spring Soil Moisture at Different Soil Depths

Table 1 shows the correlation analysis results for spring soil moisture at different depths. Both similarities and major differences are noticed in the meteorological factors affecting the spring soil moisture of the Songnen Plain and the Sanjiang Plain. For both locations, previous autumn precipitation has an impact on spring soil moisture at all soil depths. A strong positive correlation is observed in this case, i.e., a greater amount of previous autumn precipitation leads to higher spring soil moisture content across all the soil layers on both the Songnen Plain and the Sanjiang Plain. The only factor affecting the spring soil moisture on the Sanjiang Plain is previous autumn precipitation, whereas for the Songnen Plain, spring temperature and snow-cover conditions, including snow-cover duration, snow-cover onset date, and maximum snow depth, also play an effect. A strong negative correlation is present between spring soil moisture on the Songnen Plain and spring temperature, i.e., higher spring temperature leads to greater soil moisture evaporation in spring, and thus lower soil moisture content on the plain. There is a strong positive correlation between spring soil moisture on the Songnen Plain and snow-cover duration. Longer duration of snow cover leads to higher soil moisture content. A strong negative correlation exists between spring soil moisture on the plain and snow-cover onset date. An earlier onset of snow cover leads to higher soil moisture content. There is a significant positive correlation between surface soil moisture (0–10 cm) and the maximum snow depth on the Songnen Plain. This means that snow depth only affects surface soil moisture. A greater maximum snow depth means a higher amount of surface soil moisture in spring. The table also shows a significant positive correlation between soil moisture on the Songnen Plain at the 20–30 cm depth range and wind speed. This relationship is, however, purely numerical and has no practical significance.

**Table 1.** Correlation coefficient between spring soil moisture and meteorological parameters at different layers of the Songnen Plain and the Sanjiang Plain.

| Area | Layer | T | P | ST | WS | SD | PPA | SCD | MSD | SDD | SOD |
|------|-------|---|---|----|----|----|-----|-----|-----|-----|-----|
| Songnen Plain | 0–30 cm | −0.455 ** | 0.135 | −0.093 | 0.168 | 0.179 | 0.656 ** | 0.484 ** | 0.28 | −0.461 ** | 0.176 |
| | 0–10 cm | −0.420 ** | 0.283 | 0.122 | −0.013 | 0.057 | 0.628 ** | 0.523 ** | 0.395 * | −0.386 * | 0.153 |
| | 10–20 cm | −0.435 ** | 0.152 | −0.05 | 0.119 | 0.149 | 0.660 ** | 0.469 ** | 0.284 | −0.424 ** | 0.156 |
| | 20–30 cm | −0.454 ** | −0.024 | −0.309 | 0.352 * | 0.292 | 0.611 ** | 0.414 * | 0.148 | −0.511 ** | 0.195 |
| Sanjiang Plain | 0–30 cm | −0.204 | 0.139 | −0.189 | 0.122 | −0.148 | 0.595 ** | 0.229 | −0.047 | 0.013 | −0.064 |
| | 0–10 cm | −0.238 | 0.253 | −0.132 | 0.092 | −0.205 | 0.566 ** | 0.282 | 0.026 | −0.004 | −0.100 |
| | 10–20 cm | −0.169 | 0.068 | −0.200 | 0.146 | −0.119 | 0.597 ** | 0.178 | −0.097 | 0.032 | −0.043 |
| | 20–30 cm | −0.182 | 0.074 | −0.221 | 0.117 | −0.099 | 0.566 ** | 0.200 | −0.071 | 0.011 | −0.041 |

Note: "*", "**": Significance at 0.05 and 0.01 levels, daily average temperature (T) (°C), total precipitation (P) (mm), surface temperature (ST) (°C), daily average wind speed (WS) (m/s), daily average sunshine duration (SD) (h), precipitation in the previous autumn (PPA) (mm), snow-cover duration (SCD) (d), maximum snow depth (MSD) (cm), snow-cover end date (SED), and snow-cover onset date (SOD).

### 3.2.2. Factors Affecting Soil Moisture at Different Depths for Each Month of Spring

Correlation analysis on soil moisture content at different depths for each month of spring (Table 2) shows the following: (1) previous autumn precipitation impacts the soil moisture at different depths for each month of spring; (2) temperature and precipitation affect soil moisture differently for different soil depths and spring months; and (3) a significant correlation exists between spring soil moisture on the Songnen Plain and snow-cover conditions, whereas such a relationship is barely present on the Sanjiang Plain. Specifically, previous autumn precipitation has a positive correlation with soil moisture at every depth for each month of spring (except for the surface soil moisture of May), i.e., a greater amount of precipitation in previous autumn leads to higher spring soil moisture content for both the Songnen Plain and the Sanjiang Plain. On the Songnen Plain, the soil moisture at each depth shows a significant negative correlation with temperature during April and May, and a significant positive correlation with precipitation in May. On the Sanjiang Plain, the soil moisture at each depth shows a significant positive correlation with temperature and precipitation in May. On the Songnen Plain, snow-cover duration shows a significant positive correlation with soil moisture at every depth for each month of spring; snow-cover onset date has a significant correlation with soil moisture at every depth in March. For April, this impact is demonstrated in soil layers at 10–20 cm and 20–30 cm depths; snow-cover end date shows a significant correlation with surface soil moisture; maximum snow depth shows a significant correlation with April soil moisture at the depth ranges of 0–10 cm and 10–20 cm. On the Sanjiang Plain, however, a significant correlation between surface soil moisture and snow-cover duration is observed only in March. The correlation coefficient also shows that for the Songnen Plain, as months go by, snow-cover duration and snow-cover onset date affect soil moisture to a lesser extent as soil depth increases. However, these two factors continue to influence soil moisture until May. The effect of maximum snow depth and snow cover end date is mostly observed in April.

**Table 2.** Correlation coefficient between spring soil moisture and meteorological parameters at different layers in each month of the Songnen Plain and the Sanjiang Plain.

| Area | Month | Layer | T | P | ST | WS | SD | PPA | SCD | MSD | SDD | SOD |
|---|---|---|---|---|---|---|---|---|---|---|---|---|
| Songnen Plain | March | 0–10 cm | −0.282 | −0.195 | 0.062 | −0.147 | 0.145 | 0.621 ** | 0.547 ** | 0.230 | −0.390 * | −0.062 |
| | | 10–20 cm | −0.278 | −0.210 | −0.095 | 0.000 | 0.138 | 0.657 ** | 0.445 ** | 0.152 | −0.504 ** | 0.014 |
| | | 20–30 cm | −0.295 | −0.283 | −0.292 | 0.195 | 0.099 | 0.600 ** | 0.372 * | 0.037 | −0.505 ** | 0.112 |
| | April | 0–10 cm | −0.380 * | 0.267 | 0.159 | 0.089 | −0.041 | 0.507 ** | 0.479 ** | 0.471 ** | −0.314 | 0.362 * |
| | | 10–20 cm | −0.427 ** | 0.136 | −0.083 | 0.179 | 0.022 | 0.558 ** | 0.461 ** | 0.363 * | −0.343 * | 0.267 |
| | | 20–30 cm | −0.367 * | 0.010 | −0.313 | 0.410 * | 0.121 | 0.567 ** | 0.441 ** | 0.181 | −0.540 ** | 0.231 |
| | May | 0–10 cm | −0.455 ** | 0.688 ** | 0.137 | −0.084 | −0.397 * | 0.410 * | 0.239 | 0.318 | −0.247 | 0.155 |
| | | 10–20 cm | −0.384 * | 0.537 ** | 0.052 | −0.017 | −0.230 | 0.532 ** | 0.346 * | 0.303 | −0.235 | 0.216 |
| | | 20–30 cm | −0.405 * | 0.415 * | −0.112 | 0.109 | −0.070 | 0.530 ** | 0.349 * | 0.271 | −0.364 * | 0.252 |
| Sanjiang Plain | March | 0–10 cm | −0.192 | 0.094 | −0.236 | 0.096 | 0.119 | 0.543 ** | 0.352 * | −0.050 | −0.175 | −0.138 |
| | | 10–20 cm | −0.057 | 0.108 | −0.165 | 0.076 | 0.002 | 0.590 ** | 0.192 | −0.147 | −0.020 | −0.097 |
| | | 20–30 cm | −0.118 | 0.158 | −0.199 | 0.052 | −0.069 | 0.576 ** | 0.200 | −0.109 | 0.033 | −0.046 |
| | April | 0–10 cm | −0.188 | −0.070 | 0.056 | 0.040 | −0.217 | 0.501 ** | 0.275 | 0.182 | 0.122 | 0.104 |
| | | 10–20 cm | −0.092 | −0.309 | −0.095 | 0.031 | 0.027 | 0.563 ** | 0.160 | 0.004 | 0.052 | 0.053 |
| | | 20–30 cm | 0.044 | −0.316 | −0.077 | 0.052 | 0.089 | 0.489 ** | 0.173 | −0.077 | −0.078 | −0.021 |
| | May | 0–10 cm | −0.454 ** | 0.616 ** | 0.018 | 0.034 | −0.427 ** | 0.271 | 0.009 | −0.002 | 0.141 | −0.159 |
| | | 10–20 cm | −0.513 ** | 0.503 ** | −0.167 | 0.194 | −0.462 ** | 0.363 * | 0.095 | −0.050 | 0.106 | −0.020 |
| | | 20–30 cm | −0.410 * | 0.497 ** | −0.034 | −0.014 | −0.433 ** | 0.385 * | 0.152 | 0.079 | 0.106 | −0.030 |

Note: "*", "**": Significance at 0.05 and 0.01 levels, daily average temperature (T) (°C), total precipitation (P) (mm), surface temperature (ST) (°C), daily average wind speed (WS) (m/s), daily average sunshine duration (SD) (h), precipitation in the previous autumn (PPA) (mm), snow-cover duration (SCD) (d), maximum snow depth (MSD) (cm), snow-cover end date (SED), and snow-cover onset date (SOD).

### 3.3. Contribution by Snow-Cover Conditions on Spring Soil Moisture of the Songnen Plain

Using meteorological parameters with significant relevance to soil moisture at each depth across all the spring months, a multiple linear regression equation is established ($p < 0.01$) (Table 3) for the Songnen Plain. Following Equation (3), the percent contribution of each meteorological factor on the spring soil moisture content of the Songnen Plain is calculated using the normalization constant (Table 4). The conclusions that follow can be drawn based on the percent contribution. (1) Snow-cover onset date and snow-cover duration affect soil moisture for the entire spring. Compared to surface soil, a longer-term effect is detected at the deeper-layer soil. For example, the effect of snow-cover onset date on the moisture level of soil at the depth range of 20–30 cm could last until May, but the percent contribution gradually drops from 20.99% to 18% and further drops to 8.12% in May. The effect of snow-cover duration on the moisture level of soil at the depth ranges of 10–20 cm and 20–30 cm also extends until May, but the percent contribution decreases gradually. For surface-layer soil (0–10 cm), the effect of snow-cover onset date is felt until March, and the percent contribution is 11.68%. The effect of snow-cover duration lasts until April, but the percent contribution declines from 17.77% to 7.84%. (2) A comparison is made between the contribution by snow-cover onset date and snow-cover duration on soil moisture. For surface-layer soil, snow-cover duration makes a greater percent contribution than snow-cover onset date. For example, for March surface soil, the percent contribution by snow-cover onset date is 11.68%, and that by snow-cover duration is 17.66%. For deeper-layer soil, the opposite trend is observed. Taking the 10–20 cm layer as an example, the percent contributions by snow-cover onset date are 20.67% and 11.25%, respectively, for March and April, while those by snow-cover duration are 9.02% and 3.74%, respectively. (3) For a single month, the impact on soil moisture by snow-cover onset date gradually increases as soil depth increases, while that by snow-cover duration gradually decreases. Taking March as an example, the percent contributions on soil moisture by snow-cover onset date at depth ranges of 0–10, 10–20, and 20–30 cm deep are 11.68%, 20.67%, and 20.99%, respectively, and the percent contributions by snow-cover duration are 17.77%, 9.02%, and 4.92%, respectively. Maximum snow depth and snow-cover end date contribute greatly to the surface soil moisture during the snow melting, but the effect lasts only for a short period. In April, they make a large contribution (up to 10.94%) to the moisture content of just the surface soil (0–10 cm).

**Table 3.** Multiple linear regression model equation between spring soil moisture and meteorological parameters at different layers of the Songnen Plain.

| Month | Layer | Multiple Linear Regression Equation | $R^2$ | DF | SE | SE of Regression Coefficient | | | | |
|---|---|---|---|---|---|---|---|---|---|---|
| March | 0–10 cm | $y = 0.469^{**}x_{PPA} + 0.324^*x_{SCD} - 0.213x_{SOD}$ | 0.552 ** | 34 | 7.87 | PPA 0.039 | SCD 0.064 | SOD 0.159 | | |
| | 10–20 cm | $y = 0.536^{**}x_{PPA} - 0.361^{**}x_{SOD} + 0.158x_{SCD}$ | 0.604 ** | 34 | 7.12 | PPA 0.035 | SOD 0.144 | SCD 0.058 | | |
| | 20–30 cm | $y = 0.497^{**}x_{PPA} - 0.388^{**}x_{SOD} + 0.091x_{SCD}$ | 0.528 ** | 34 | 8.87 | PPA 0.044 | SOD 0.179 | SCD 0.072 | | |
| April | 0–10 cm | $y = 0.370^*x_{PPA} + 0.285x_{SDD} + 0.240x_{MSD} + 0.172x_{SCD} - 0.012x_T$ | 0.491 ** | 31 | 6.33 | PPA 0.031 | SDD 0.104 | MSD 0.071 | SCD 0.062 | T 0.863 |
| | 10–20 cm | $y = 0.408^{**}x_{PPA} - 0.285x_T - 0.248x_{SOD} + 0.098x_{MSD} + 0.083x_{SCD}$ | 0.508 ** | 31 | 5.23 | PPA 0.026 | T 0.63 | SOD 0.109 | MSD 0.059 | SCD 0.054 |
| | 20–30 cm | $y = 0.377^{**}x_{PPA} - 0.369^{**}x_{SOD} - 0.243x_T + 0.175x_{WS} + 0.125x_{SCD}$ | 0.629 ** | 31 | 5.77 | PPA 0.029 | SOD 0.132 | T 0.69 | WS 1.853 | SCD 0.053 |
| May | 0–10 cm | $y = 0.715^{**}x_P + 0.313^{**}x_{PPA} - 0.204x_T + 0.178x_{SD}$ | 0.643 ** | 32 | 4.67 | P 0.047 | PPA 0.022 | T 0.655 | SD 0.036 | |
| | 10–20 cm | $y = 0.439^{**}x_P + 0.427^{**}x_{PPA} - 0.134x_T + 0.132x_{SCD}$ | 0.559 ** | 32 | 4.57 | P 0.034 | PPA 0.023 | T 0.636 | SCD 0.037 | |
| | 20–30 cm | $y = 0.408^{**}x_{PPA} + 0.283^*x_P - 0.185x_{SOD} - 0.176x_T + 0.100x_{SCD}$ | 0.507 ** | 31 | 4.72 | PPA 0.024 | P 0.035 | SOD 0.097 | T 0.667 | SCD 0.039 |

Note: "*", "**": Significance at 0.05 and 0.01 levels.

**Table 4.** The percent contributions on soil moisture by meteorological parameters at different layers of the Songnen Plain (%).

| | March | | | April | | | May | | |
|---|---|---|---|---|---|---|---|---|---|
| Layer (cm) | 0–10 | 10–20 | 20–30 | 0–10 | 10–20 | 20–30 | 0–10 | 10–20 | 20–30 |
| PPA | 25.75 | 30.71 | 26.89 | 16.82 | 18.46 | 18.41 | 14.26 | 21.09 | 17.93 |
| SOD | 11.68 | 20.67 | 20.99 | - | 11.25 | 18.00 | - | - | 8.12 |
| SCD | 17.77 | 9.02 | 4.92 | 7.84 | 3.74 | 6.10 | - | 6.51 | 4.42 |
| MSD | - | - | - | 10.94 | 4.43 | - | - | - | - |
| SDD | - | - | - | 12.97 | - | - | - | - | - |
| T | - | - | - | 0.54 | 12.92 | 11.84 | 9.32 | 6.61 | 7.76 |
| WS | - | - | - | - | - | 8.56 | - | - | - |
| P | - | - | - | - | - | - | 32.61 | 21.69 | 12.46 |
| SD | - | - | - | - | - | - | 8.11 | - | - |

## 4. Discussion

(1) In the past, researchers have studied the effect of snow cover on spring soil moisture. The results show that snow-cover promotes the change rate of soil moisture [29] and has a noticeable impact on shallow-layer soil moisture [23]. Greater snow-cover depth and longer snow-cover duration tend to have a more significant and longer-term impact on shallow-layer soil moisture content [16]. Pan et al. [30] suggested and tested an empirical approach to estimated root-zone soil moisture in snow-dominated regions using a soil moisture diagnostic equation that incorporates snowfall and snowmelt processes. The result indicated that the soil moisture diagnostic equation is capable of accurately estimating soil moisture in snow-dominated regions after the snowfall and snowmelt processes are included in the soil moisture diagnostic equation. Qi et al. [19] have investigated snow performs similar to an important reservoir. In March–May, the soil moisture would decrease at least 20.1% when there is no snow, and the main cropland area suffers more. Shinoda et al. [16] found that the yearly maximum snow depth represents a major portion of the soil water upon snow disappearance. Potopova et al. [31] presented a detailed analysis which showed that snow-cover characteristics can significantly influence soil water saturation during the first part of the growing season, while seasonal amount of SWE can explain up to 45% of soil moisture variability during early summer (April–May–June). Liang [32] investigated the farmland of Northeast China and concluded that snow-cover depth has a strong positive correlation with April and May soil moisture. This correlation is, however, spatially differentiated, with regions showing a significant correlation concentrated mostly in the southwestern part of Heilongjiang Province. A greater correlation was also observed between snow-cover depth and April soil moisture by Liang et al. [30], which is in line with the conclusion of this paper. Taking the Songnen Plain and the Sanjiang Plain in Heilongjiang Province as examples, this study analyzes the impact of snow cover on spring soil moisture. It is noticed that snow cover plays a greater effect on the spring soil moisture content of part of the Songnen Plain located in northwestern Heilongjiang Province. This effect could last until May, but it impacts April soil moisture greatly. Yet, snow cover has less effect on the spring soil moisture content of the part of the Sanjiang Plain located in southeastern Heilongjiang Province. Our results show that previous autumn precipitation, snow-cover duration, and snow-cover onset date are the most important factors affecting the soil moisture of each layer on the Songnen Plain during the spring months. Due to the winter soil freezing, the precipitation of the previous autumn is well contained with little loss. Snow cover also acts as an insulation layer, conserving soil moisture [5]. An earlier onset of snow cover enhances the conservation effect on soil moisture. Soil memory ensures that previous autumn precipitation and snow cover could have a longer-term effect on soil moisture conservation at deeper layers, lasting until May. Snow-cover depth and snow-cover end date mainly affect the April soil moisture content at shallow layers. Our analysis shows that snow melting mainly takes place from the end of March to early April. During this time, the seeping of snowmelt has a great impact on soil moisture in the shallow layers. Greater snow-cover depth and delayed snow-cover end date lead to higher

shallow-layer soil moisture. In May, as the temperature rises and precipitation increases, the effect of snow cover on soil moisture gradually decreases.

(2) The effects of snow cover on soil moisture were also different in different study areas. Douville [20] suggested that the effect of spring snowmelt on soil moisture can last until summer. McNamara [21] found it lasts until late spring. Although Zhang et al. [22] reported that snow cover has an impact on soil moisture mainly as it melts. Our analysis points to clear differences in the effect played by snow-cover conditions for soil moisture of the Songnen Plain and the Sanjiang Plain. A further comparison is made on the climatic background of both locations. Five meteorological factors (spring temperature, precipitation, surface temperature, wind speed, and sunshine duration) are selected for comparison. The results are shown in Table 5. Variance analysis reveals significant differences in the effect of four of these meteorological factors, except for spring temperature, i.e., compared to the Sanjiang Plain, the Songnen Plain experiences significantly less spring precipitation, lower surface temperature, greater wind speed, and longer sunshine duration. In particular, the spring precipitation on the Songnen Plain is only about 70% that of the Sanjiang Plain, which means a drier climate for the former. It can thus be concluded that snow helps to conserve and replenish soil moisture in dry areas. For places of higher humidity, the contribution of snow cover to soil moisture is not as significant.

**Table 5.** Statistical table of spring climate conditions of the Songnen Plain and the Sanjiang Plain for 1983–2019 (average value and variance analysis result).

|  | **Songnen Plain** | **Sanjiang Plain** |
|---|---|---|
| T | 5.54 °C [a] | 5.40 °C [a] |
| P | 69.48 mm [a] | 99.16 mm [b] |
| ST | −2.88 °C [a] | −1.86 °C [b] |
| WS | 3.63 m/s [a] | 3.38 m/s [b] |
| SD | 745.1 h [a] | 677.7 h [b] |

Note: "a", "b": Variance analysis result between the Songnen Plain and the Sanjiang Plain.

(3) Among all the snow-cover parameters, snow-cover duration, snow-cover onset date, maximum snow depth, and snow-cover end date are thought to have a greater contribution to soil moisture on the Songnen Plain. The variation characteristics of these four snow-cover parameters have been analyzed for the Songnen Plain, and they are shown in Figure 6. The average value of snow-cover duration is 104 d, the value of maximum snow depth is 13.6 cm, the average snow-cover onset date is November 1, and the average snow cover end date is April 2. On the Songnen Plain snow-cover duration showed declining trend, and the decline rate was −0.19 d/a while the trend was not significant. The increase rate maximum snow depth was 0.20 cm/a, which was significant ($p < 0.05$). Snow-cover onset date showed increasing trend ($p < 0.05$), and the increase rate was 0.36 d/a. Snow-cover end date showed declining trend ($p < 0.05$), and the decline rate was −0.41 d/a. Compared with those in the 1980s, maximum snow depth on Songnen Plain increased by 8.0 cm, snow-cover onset date was 14.4 days later, and snow-cover end date was 16.4 days earlier in the 2010s. Changes in any of these snow-cover parameters will negatively impact the maintenance of soil moisture. Despite the significant increase in snow-cover depth, this factor only influences the April surface soil and no noticeable influence on the deeper layers.

(4) The soil become "wet soil" due to the snow melting, which keeps snow signal for a long time and interacts with the atmosphere in the long term [33]. This soil memory can influence regional and even global climate change [34,35]. Previous studies have not studied the spatial difference of snow-cover influence, but this study result indicates that the impact of snow cover on soil moisture is different in different regions, which means that the indirect impact of snow cover on climate is different when the study area changes. In the future, more attention should be focus on the difference of snow effect in different regions. The cause of this difference also needs further research, whether it is caused by the

difference of soil properties or climate background. For agriculture, exploring the influence of snow cover on spring soil moisture can improve the accuracy of soil moisture estimation in spring and predict crop growth. Based on the estimated soil moisture, more efficient irrigation scheme can be developed, and water resources can be rationally allocated.

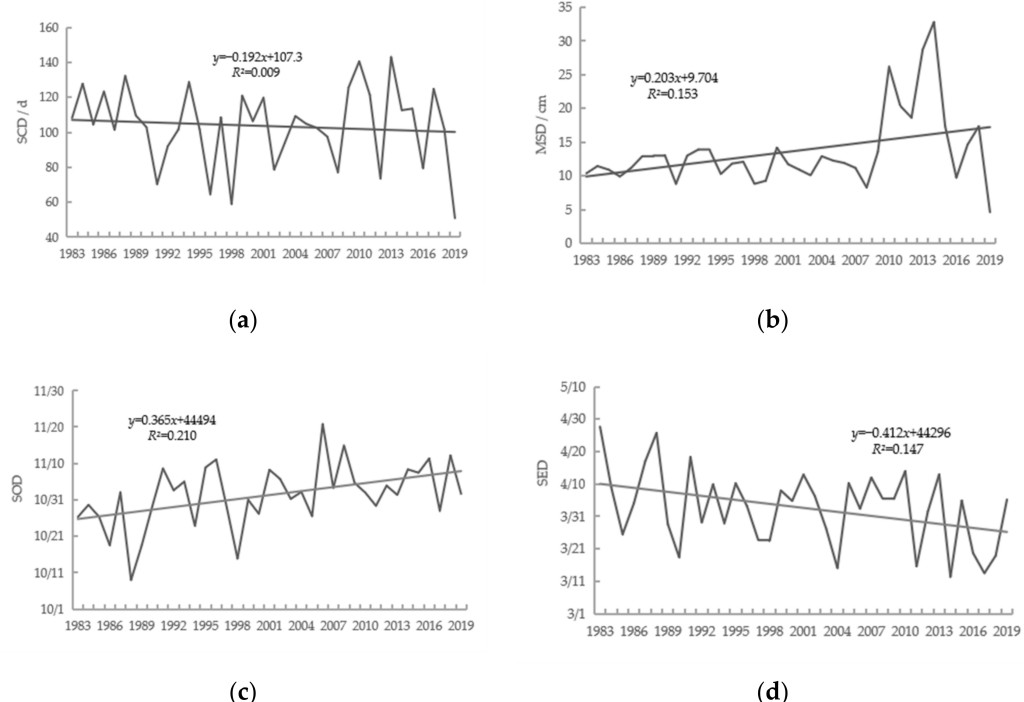

**Figure 6.** Interannual variation of snow-cover parameters for Songnen Plain from 1983 to 2019. ((**a**): SCD, (**b**): MSD, (**c**): SOD, (**d**): SED).

## 5. Conclusions

During the period of 1983–2019, the average spring soil moisture contents for the Songnen Plain and the Sanjiang Plain are 81.39% and 92.37%, respectively. Compared to the Sanjiang Plain, the Songnen Plain has significant lower spring soil moisture content and greater interannual variation of soil moisture. The Songnen Plain has a significantly lower spring soil moisture content than the Sanjiang Plain across all soil layers for the spring months.

Among all the meteorological factors, previous autumn precipitation is the main influencer of the spring soil moisture content of both the Songnen Plain and Sanjiang Plain. Snow-cover conditions have little effect on the spring soil moisture content of the Sanjiang Plain, but affects that of the Songnen Plain greatly. For the Songnen Plain, snow-cover duration and snow-cover onset date both correlate significantly with soil moisture across all the spring months. The percent contribution on soil moisture by snow-cover duration and snow-cover onset date is about 30% for March. As the months go by, the percent contribution gradually decreases. The impact on the shallow soil layer disappears in May, but a contribution of 12% is still felt at deeper layers (20–30 cm). The maximum snow depth and snow-cover end date only affect April surface soil moisture for a short while, but the percent contribution is as high as 24%.

Comparing the climate characteristics of the Songnen Plain and the Sanjiang Plain, the former is found to have a drier climate, while the different impacts snow cover has on soil moisture of the two areas could come from the differences in their climatic conditions. Snow has a stronger soil moisture conservation effect for drier areas.

Analyzing the variation characteristics of snow-cover parameters in the Songnen Plain from 1983 to 2019, it is found that the average maximum snow depth is found to increase greatly, along with greatly delayed snow-cover onset date and much earlier snow-cover

end date. Snow-cover duration, however, does not change significantly. In terms of the change rate, changes in snow-cover end date and onset date happen more rapidly, and the increase in maximum snow depth happens more slowly. Overall, changes in snow-cover conditions intensify the decrease in spring soil moisture content on the Songnen Plain, which may lead to reduced grain production.

This study is only limited to two agricultural bases in Heilongjiang Province and does not conduct a detailed analysis of all the stable snow covers regions. The conclusions may have regional limitation. It has not conducted an in-depth study on the mechanism that the different impacts snow cover has on soil moisture of the two areas with different climatic conditions. It will be subject to special research in the future.

**Author Contributions:** M.P. analyzed the data and drafted the manuscript; F.Z., J.M. and L.Z. completed the manuscript and made major revision; J.Q. and L.X. searched references; Y.L. checked and proofread the manuscript. All authors have read and agreed to the published version of the manuscript.

**Funding:** This research was funded by National Natural Science Foundation of China under grant number 41771067, which is Key Projects of Natural Science Foundation of Heilongjiang Province (No. ZD2020D002). This research was also funded by National Natural Science Foundation of China under grant number 41665007, which is supported by open-ended fund of the Shenyang Institute of Atmospheric Environment of China Meteorological Administration (No. 2021SYIAEKFMS28).

**Institutional Review Board Statement:** Not applicable.

**Informed Consent Statement:** Not applicable.

**Data Availability Statement:** Not applicable.

**Conflicts of Interest:** The authors declare no conflict of interest.

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
