# Peer review of "Effect of Snow Cover on Spring Soil Moisture Content in Key Agricultural Areas of Northeast China"

_sustainability, doi:10.3390/su14031527_

Round 1

Reviewer 1 Report

The impact of snow cover on spring soil moisture is analyzed quantificationally for two agricultural bases of Heilongjiang Province in this article. The climatic background of both study sites was different, and the results are opposite. This can provide important theoretical support for local adapting and mitigating strategies. The method is described clearly and the results are of interest. 

 It has merits, but is not suitable for publication as it currently stands. Therefore, my decision is "Major Revision." 

 Specific comments

Point 1 The authors should outline the main objectives of the research in the introduction.

Point 2 p. 5, li. 196 ‘R’ should be italic.

Point 3 Table 1-3. What asterisk means?

Point 4 p. 10, li. 340. Equation (4) is not exist.

Point 5 it should be small p... please apply in the entire text.

Point 6 please elaborate limitations and possible new orientation for future studies... (Conclusion).

Point 7 please check the references based on the format.

Reviewer 2 Report

  • Introduction is a pile of literature. What is the significance of this study? What scientific or technical problems do you want to solve?
  •          Please lso write the objectives of this study clearly?.
  • -          Line 151. Please change Study methods to data analysis?

  • Line 163. Why authors selected Kriging method? I agree that it is one of the most applicable interpolation methods but in some cases, different methods are better, for example IDW  Did authors checked different methods to select the most applicable of them? How many sample were used as training stations and how many as testing? Which method was selected to select testing samples? Please add this information.
  • Please increase the resolution of figures 2,3 and 4?.
  • What are stars (*,**) mean in tables 1, 2 and 3?. Please add the explanations under the tables?.
  • In table 3 and 4, did you use multiple linear regression model or Stepwise multiple linear regression?

-          What are the letters a and b mean in table 5?. Please add the explanation under the table.  

-          The discussion is weak and it lacks from the references. Please rewrite the discussion according to the order of results.

-          Line 387.  Liang et al. please write the number of the reference after et al.

-          How would the outcomes of work directly contribute to global climate change mitigation? What are the likely research impacts of this work globally, nationally and locally? How would this research work advance the previous work done in the existing field of study and/or across other fields? Could you please add new section before conclusion to write about them?

Reviewer 3 Report

Effect of Snow Cover on Spring Soil Moisture Content in Key Agricultural Areas of Northeast China

In this paper, the authors have investigated the effect of snow cover in selected meteorological stations in northeast China. The following are the comments that need the kind attention of the author for improvement.

  1. The abstract and the introduction sections are presented nicely. Line 18 shows 19 stations whereas Line 89 shows 18 stations. Kindly check and rectify the error. Addition of hypothesis, contribution of this paper to the existing literature/research gap and how this study is different from the others would strengthen the quality of article.
  2. Line 97: Standard unit shall be used to indicate the area like million hectares.
  3. Line 115: list of meteorological stations can be listed in the main text or as a footnote
  4. Line 123: There is no information on the frequency of data collected spanning 1983–2019. Is it daily or weekly etc.?
  5. Line 128: [] Is it a missing reference?
  6. Line 140: Mention RSH as relative soil humidity when used as an acronym
  7. Line 154: change determined to determine
  8. Line 191: change R to R2
  9. In Table 1 and Table 2, the details of the meteorological parameters shall be given at the bottom of the table for easy reference
  10. Degrees of freedom, significance of regression coefficients along with the standard error have to be reported in Table 3
  11. Table 4 presentation is not clear. Table 3 and 4 title is same and they need to be presented following a standard format to show the estimates of the multiple regression model
  12. Table 5: Which statistical test was used? The title needs more clarity
  13. The paper has used multiple statistical tools to analyze the data without describing much on the theoretical background. For instance, a multiple linear regression model has been used to capture the relationship between soil moisture and meteorological parameters. While looking into the correlation, there was a low association in terms of linearity. Did the authors check for non-linearity and assumptions of the regression model? An explanation is required on this aspect.
  14. The results are presented very well with in-depth discussion but lack sufficient literature evidence.
  15. The conclusion section needs to be improved by adding the implications for crop production from the research findings. Also, the limitation of this study and future directions of research shall be given.

Round 2

Reviewer 2 Report

the authours must answer about last point

Point 11 How would the outcomes of work directly contribute to global climate change mitigation? What are the likely research impacts of this work globally, nationally and locally? How would this research work advance the previous work done in the existing field of study and/or across other fields? Could you please add new section before conclusion to write about them?

Reviewer 3 Report

Glad to see the revised version of the manuscript and the paper is much improved over its previous version. I agree with all the revision comments with some reservations in the information provided in the revised Table 3. 

The added standard error and degrees of freedom values need more clarification. In multiple regression estimation, every partial regression coefficient will have a standard error (SE). But in Table 3, only one SE has been mentioned for each regression equation. Second, the degrees of freedom (DF) is too low for the presented regression models. The DF is computed as n-k-1, where 'n' is the number of observations, and 'k' is the number of variables. In this case, what is the number of observations considered for each regression model? For eg., regression pertaining March, 0-10 cm has a DF of 3 which means n-k-1=3. Here 'k' is 3, so 'n' should be 7. Is it? The authors are requested to explain this. Usually, 12 DF is suggested (a thumb rule) for better estimates. Also, how 55 DF is arrived for the last equation (May, 20-30cm)
